# Dietary Intake of Capsaicin and Its Association with Markers of Body Adiposity and Fatty Liver in a Mexican Adult Population of Tijuana

**DOI:** 10.3390/healthcare11223001

**Published:** 2023-11-20

**Authors:** Yesenia Martínez-Aceviz, Ana Alondra Sobrevilla-Navarro, Omar Ramos-Lopez

**Affiliations:** 1Faculty of Medicine and Psychology, Autonomous University of Baja California, Tijuana 22390, Baja California, Mexico; yesenia.guadalupe.martinez.acevis@uabc.edu.mx (Y.M.-A.); ana.sobrevilla@academicos.udg.mx (A.A.S.-N.); 2Department of Biomedical Sciences, University Center of Tonalá, University of Guadalajara, Guadalajara 44100, Jalisco, Mexico

**Keywords:** dietary capsaicin, adiposity, fatty liver, Mexican population

## Abstract

**Background:** Capsaicin (CAP) is the main chemical component responsible for the pungency (burning pain) of the chili plant (*capsicum* spp.), whose metabolic functions include energy balance and fatty acid oxidation. The aim of this study is to analyze the association of dietary capsaicin consumption with markers of adiposity and fatty liver in a Mexican adult population. **Methods**: This cross-sectional/analytical study recruited 221 subjects aged 18 to 65 years who were resident in the city of Tijuana, Baja California, Mexico. The daily CAP intake was analyzed through a validated chili/CAP consumption questionnaire. Anthropometric and biochemical measurements were performed following standardized protocols. Adjusted Pearson’s correlations were applied to analyze the association of CAP with adiposity and fatty liver markers. **Results:** In this study, the daily average consumption of CAP was 152.44 mg. The dietary CAP consumption positively correlated with BMI (r = 0.179, *p* = 0.003), hip circumference (r = 0.176, *p* = 0.004) and body adiposity index (r = 0.181, *p* = 0.001. Likewise, the daily CAP intake positively correlated with hepatic steatosis index (r = 0.158, *p* = 0.004), fatty liver index (r = 0.141, *p* = 0.003) and lactate dehydrogenase (r = 0.194, *p* = 0.016) after statistical settings. **Conclusions**: The results of this study suggest positive associations between dietary CAP consumption and the markers of body adiposity and fatty liver in a Mexican adult population.

## 1. Introduction

Chili is a distinctive culinary ingredient of Mexican culture, which is commonly consumed directly in fresh state (free of industrial processing or additives) or as part of dishes prepared with both fresh and dried chilies in daily or festive cooking preparations [1]. These have a long history of consumption to impart flavor, color and preserve foods, as well as a series of uses in the pharmaceutical industry and traditional medicine [1]. Due to its bitter and spicy flavor, chili has played an important role in Mexican gastronomy since pre-Hispanic times and is typically used as a condiment or to accompany a variety of meals [2]. These fruits are available as a natural resource in a wide range of shapes, sizes, colors and flavors, as they belong to the genus *Capsicum* spp., which is part of the Solanaceae family and has five domesticated species: *C. annuum*, *C. chinense*, *C. frutescens*, *C. baccatum* and *C. pubescens* [3]. The chilies most consumed in Mexico are “jalapeño, serrano, habanero, de arbol, güero, manzano, chilaca, poblano and bell pepper”, while the dried chilies are “piquín, catarino, pasilla, morita, chipotle, ancho, guajillo and mulatto” [4]. These chilies are widely consumed in Mexico fresh or as ingredients in a variety of sauces/dips to accompany the dishes. At present, several indigenous groups in Mexico have safeguarded traditional/local environments related to the production of chili [5]. Despite the use of chili as a food or cooking ingredient emerging in the Americas approximately 500 years ago, at present, this plant has spread to Asian, African, European and American countries due to commercial and global cultural processes [1]. Also, the food industry has used chili derivatives to produce other products attributable to the flavor, color and bactericide properties of this plant, which are produced on a large scale and distributed worldwide [1]. In addition to its use as a common and important spice-enhancing taste, chili is also applied for medicinal purposes in pain-relief, as a potential effective anesthetic agent, and in the treatment of various gastrointestinal complains [1].

Capsaicin (CAP) is an extremely volatile, odorless and hydrophobic compound responsible for the spiciness of the chili plant [5]. The sensory response caused by CAP is known as pungency, a chemical irritation that results from stimuli of the trigeminal nerve, being responsible for producing the sensations of heat and pain in the mouth [5]. The amount of capsaicin varies depending on the type of crop, ripening, location, climate and fruit of the chili, whose largest volume is located in the placenta, except for red, yellow and green peppers, which do not contain CAP [6]. CAP is absorbed by the digestive tract, particularly the jejunum, ileum and duodenum, thus transporting capsaicin to the portal vein [7]. CAP participates in many physiological processes, including pain transduction, immune response, glucose metabolism, cardiovascular function, thermogenesis and satiety [8,9,10,11]. Several targets have been implicated in these functions, including the activation of several signaling pathways, including AMP-activated protein kinase/AKT, calcium transduction, transient receptor potential vanilloid 1 (TRPV1), peroxisome proliferator-activated receptor α (PPAR-alpha), uncoupling protein 1 (UCP1) and glucagon-like peptide 1 (GLP1) [8,9]. Moreover, the modulation of the abundance and composition of gut microbiota as well as nociceptive sensory neurons excitation have been identified as potential mechanisms underlying the physiological effects of CAP [10].

In this context, epidemiological studies have evaluated the association between chili/capsaicin consumption and adiposity, with controversial results. A prospective study reported an inverse association between chili intake and the risk of being overweight/obesity in Chinese adults [12]. In fact, it was observed that the regular consumption of capsaicinoid compounds may reduce abdominal adiposity, appetite and energy intake probably via the stimulation of the TRPV1 receptor [13]. Meanwhile, another trial reported that the acute administration of CAP was able to increase resting energy expenditure without subjacent changes in energy intake, appetite and blood levels of orexigenic/anorexigenic peptides in obese adolescents and young adults [14]. Accordingly, the acute administration of CAP promoted energy expenditure and fat oxidation in apparently healthy individuals [15]. The intraduodenal infusion of CAP significantly increased satiety in healthy volunteers without affecting the release of satiety hormones [16]. Moreover, a systematic review and meta-analysis suggested that the daily consumption of capsaicinoids may contribute to body weight management [17]. Moreover, a recent meta-analysis stated that CAP supplementation may have modest effects on reducing body weight, BMI, and WC in overweight/obese individuals [18]. On the contrary, an opposite effect was detected in another Chinese sample, where the consumption of spicy food was positively associated with adiposity measures (especially central obesity) in males, but not in females [19]. Moreover, increased risks of being overweight and obese were observed in Chinese people who frequently consumed chili [20]. Other studies on Asians similarly evidenced positive associations between a spicy taste and the consumption of chilies and spicy foods with general and abdominal obesity levels [21,22,23]. Indeed, a meta-analysis of the available cross-sectional studies revealed an increased risk of being overweight/obese in individuals in the largest category of spicy food intake compared to those in the smallest category [24]. Thus, further studies are needed to elucidate the association of CAP with body adiposity and metabolic features, taking into account the characteristics of the study populations as well as the cultural and related environmental factors of each region.

In Mexico, there is a high prevalence of obesity, where more than 70% of the total population are overweight or obese. Indeed, Mexico is currently the second most obese country in the world [25]. Moreover, liver cirrhosis is the fourth leading cause of morality among Mexicans, being the result of excessive alcohol consumption, viral hepatitis, and obesity-related fatty liver, the main causes of liver disease in the country [26]. Despite disease burden, no effective strategies have been implemented in Mexico to control obesity and liver disease burden to date. In addition, no available studies assess the effects of CAP intake on obesity and metabolic syndrome in Mexican individuals. Exploring these relationships can help to elucidate the role of chili/CAP in obesity status in Mexico to establish personalized nutritional recommendations based on chili/CAP intake. Therefore, the aim of this study is to analyze the association of dietary CAP consumption with markers of adiposity and fatty liver in a Mexican adult population.

## 2. Materials and Methods

### 2.1. Participants

This cross-sectional/analytical study included a general sample of about 223 Mexican adult subjects, aged 18–65 years old, and from both genders. Apparently, healthy (by self-report) participants were randomly recruited from the “Centro de Atención Integral de la Salud (CAIS)” at the Faculty of Medicine and Psychology of the Autonomous University of Baja California (UABC), in the city of Tijuana, Baja California, Mexico. The recruitment process consisted of disseminating an invitation to participate in the study using the social media available at the UABC. Major exclusion criteria included a clinical history of diabetes, cardiovascular diseases (hypertension, coronary heart disease and cerebrovascular disease), thyroid disorders, ulcerative colitis, dyspepsia or related gastric problems; pregnant or lactating women; smokers; individuals with chronic sinus problems; subjects taking any prescribed medication that might affect their taste perception and blood lipid/glucose levels; and those who reported consuming a special diet that restricted nutrients or calories in the last three months prior to the study. Alcohol drinkers (consuming more than 20 g/d and 40 g/d of ethanol in women and men, respectively) were also excluded. Two individuals had diagnoses of hypothyroidism and were excluded. Thus, a total of 221 subjects were finally included in the study.

The study protocol was presented to the Research Ethics Committee of the UABC for approval (code: D235, accepted on 22 October 2019). Additionally, the research was conducted in accordance with the ethical principles for human research according to the Declaration of Helsinki [27]. Moreover, the participants were informed about the details and procedures of the protocol, who voluntarily provided written informed consent.

### 2.2. Anthropometric Measurements and Blood Pressure

A stadiometer (Rochester Clinical Research, New York, NY, USA) was used to estimate the height (cm) of the volunteers. Body weight (kg) and total body fat (%) were measured using a Tanita SC-331S (body composition analyzer, Tanita Corporation, Tokyo, Japan). Body mass index (BMI) was calculated by dividing body weight by squared meters (kg/m^2^). Waist circumference (WC) was determined between the lower rib and top of the iliac crest, whereas hip circumference (HC) was measured over the great trochanters. A stretch-resistant tape was used to collect both circumferences. Waist to hip ratio (WHR) was calculated by dividing the WC by the HC. All anthropometric measurements were collected by trained nutritionists in a conditioned/equipped clinic in CAIS. The body adiposity index (BAI) was calculated as BAI = [hip circumference (cm) ÷ height (m) 1.5] − 18 [28], whereas the visceral adiposity index (VAI) was estimated as VAI = (WC (cm)/(39.68 + (1.88 × BMI))) × (triglycerides/1.03) × (1.31/high-density lipoprotein cholesterol or HDL-c) for males and VAI = (WC (cm)/(36.58 + (1.89*BMI))) × (triglycerides/0.81) × (1.52/HDL-c) for females [29]. The BAI is a useful measure of body fat percentage, which can be calculated from the HC and height only, with an accessible application in a clinical setting [28]. Furthermore, the BAI has been validated in individuals of Mexican ancestry with widely varying adiposity levels [28]. Likewise, the VAI has been identified as a reliable parameter reflecting abdominal fat distribution and a promising tool to identify metabolic syndrome and cardiometabolic risk [29]. This index also uses conventional issues for its calculation, being simple to apply in epidemiological research. Both the BAI and VAI indices were calculated as complements of other conventional measurements of adiposity, such as BMI and fat percentage.

Systolic blood pressure (SBP) and diastolic blood pressure (DBP) were determined by triplicate following the standardized criteria using an automated sphygmomanometer [30].

### 2.3. Dietary Intake and Appetite

Habitual dietary intake was assessed using 24 h recalls, which were validated in the Mexican population [31,32]. Each subject was asked about the type, amount, and mode of preparation of all foods consumed during two weekdays and one weekend day using food scales as models. Dietary records were revised and coded by a trained dietitian using the Nutrikcal computer program (Nutrikcal VO^®^, Mexico City, Mexico) in order to obtain the average daily intakes of energy/calories and macronutrients (carbohydrates, fats and proteins). Moreover, a validated semi-quantitative food frequency questionnaire (FFQ) was used for estimating the CAP consumption that included the most consumed types of chilies and the dishes prepared with chili in Mexico [4,33]. The participants were asked how often they consumed chili during the past three months according to the following categories: never; monthly; weekly and daily [4,33]. This FFQ included standardized quantitative chili servings to estimate the portion size consumed either in household measures or grams. The frequency of the consumption of pre-defined portions of chilies and dishes prepared with chilies was further multiplied by the corresponding CAP content of each chili depending on fresh or dried forms [4,33]. The total amount of CAP intake of each subject was estimated by adding up the CAP content for the daily reported consumption of each chili and chili dish [4,33]. Then, the subjects were stratified according to the following CAP intake categories: occasional CAP consumers and daily CAP consumers (1–100 mg/d; 100–400 mg/d; and 400–900 mg/d) for comparison purposes. Furthermore, appetite was assessed using the simplified nutritional appetite questionnaire (SNAQ), a four-item tool comprising self-assessments of appetite, satiety, taste of food and number of meals per day [34]. The total score was used as a continuous variable in additional analyses (range from 4 to 20) [34].

### 2.4. Blood Tests

Overnight fasting blood samples were drawn and properly centrifuged for further serum processing. Serum glucose, total cholesterol, triglycerides, HDL-c, total proteins, albumin, globulins, creatinine, lactic dehydrogenase (LDH), aspartate ami-notransferase (AST), alanine aminotransferase (ALT), gamma glutamyl transpeptidase (GGT), total bilirubin (TB), direct bilirubin (DB), alkaline phosphatase (ALP), uric acid, total calcium, total phosphorus, total iron, creatine kinase, sodium, potassium and chlorine were determined using appropriate commercial kits and read on the Mindray BS-200 analyzer (Mindray Medical International Limited, Shenzhen, China). Low-density lipoprotein cholesterol (LDL-c) was calculated according the Friedewald formula [35]. The fatty liver index (FLI) [36] and hepatic steatosis index (HSI) [37] were fitted as proxies of steatotic liver disease as follows: HIS = 8 × (ALT/AST) + BMI + 2 (if type 2 diabetes) + 2 (if female); FLI = (e(0.953 × ln(triglycerides) + 0.139 × BMI + 0.718 × ln(GGT) + 0.053 × WC − 15.745))/(1 + e(0.953 × ln(triglycerides) + 0.139 × BMI + 0.718 × ln(GGT) + 0.053 × WC − 15.745) × 100. Moreover, the triglyceride to glucose index (TyG) mirroring insulin resistance [38] was estimated as follows: ln (fasting triglycerides [mg/dL] × fasting glucose [mg/dL]/2). In addition, the following indices were calculated as markers of inflammation using simple divisions between the following parameters as absolute values: neutrophil to lymphocyte ratio (NLR); platelet to lymphocyte ratio (PLR); eosinophil to basophil ratio (EBR); eosinophil to lymphocyte ratio (ELR) and lymphocyte to monocyte ratio (LMR) [39,40].

### 2.5. Statistical Analyses

The normality values of the main study variables (CAP intake, body adiposity markers and metabolic parameters) were verified by Kolmogorov–Smirnov tests. Continuous variables were expressed as means ± standard deviations, and categorical variables as number of cases. Statistical differences in the anthropometric and biochemical variables between CAP consumption categories were estimated using ANOVA tests. Post hoc (Bonferroni) tests were applied to determine specific differences between the study groups. Pearson’s correlations were used to analyze the association between CAP consumption and markers of body adiposity and fatty liver, which were adjusted for confounding variables, including sex, age, energy intake and appetite. Statistical analyses were performed using the statistical programs IBM SPSS 20 (IBM Inc., Armonk, NY, USA). Statistical significance was set to *p*-value ˂0.05.

## 3. Results

Table 1 shows the demographic, anthropometric and clinical characteristics of consumers and non-consumers of CAP, where it can be observed that those who consume CAP daily have a larger HC (*p* = 0.032) and a higher BAI (*p* = 0.022) compared to those who do not consume CAP (Table 1).

The nutritional characteristic analyses showed no significant differences concerning appetite, total calories and percentages of protein, fat and carbohydrates according to CAP consumption (Table 2).

The metabolic characteristics (biochemical parameters of liver function, kidney status, glucose and lipid profiles) of the population categorized by CAP evidenced higher serum levels of LDH (*p* = 0.030) and TB (*p* = 0.019) in CAP consumers than their counterparts, whereas no significant differences were found for the rest of the markers (Table 3).

Table 4 shows the inflammatory profiles of consumers and non-consumers of CAP. It can be observed that there are no differences between both groups (Table 4).

The correlations between CAP consumption and anthropometric parameters are plotted in Figure 1. CAP intake positively correlates with BMI (r = 0.179, *p* = 0.003), HC (r = 0.176, *p* = 0.004) and BAI (r = 0.181, *p* = 0.001) after adjustments by age, sex, appetite, calories, protein, fat and carbohydrate ingestion levels.

Figure 2 shows the correlations between CAP intake and metabolic markers of liver function. Daily CAP consumption positively correlates with HSI (r = 0.158, *p* = 0.004), FLI (r = 0.141, *p* = 0.003) and LDH (r = 0.194, *p* = 0.016) after statistical settings (Figure 2).

## 4. Discussion

CAP is a spicy reactive compound with implications on health status [41]. The findings of this study suggest modest but significant positive associations between CAP intake and markers of adiposity (BMI, BAI and HC) and fatty liver (HSI, FLI and LDH) in the adult Mexican population of Tijuana. These findings may help in the development of personalized nutritional intervention strategies for obesity and liver diseases based on CAP consumption levels. However, since the absolute correlation rates are moderate, additional studies are needed to validate our results, taking into consideration CAP dosage, as well as to explore the additional relationships between CAP consumption and adiposity/metabolic status in other Mexican regions and worldwide.

The most important source of CAP comprises a variety of chilies, but its consumption varies between regions due to eating, cultural and social differences [42]. To date, there is scarce evidence concerning the amount of CAP consumed by the Mexican population. In this study, a high consumption level of CAP was found among participants living in Tijuana (Northwest Mexico), with an average of 152.44 mg per day. Of note, lower CAP levels were reported in different Mexican areas, including Puebla (31.99 mg/d), Mexico City (29.84 mg/d) and Yucatán (24.54 mg/d), located at the Center and South of Mexico. Moreover, a study conducted in Korea estimated that the maximum consumption level of CAP was 78.67 mg/d [43]. Therefore, further studies are required throughout the Mexican territory and worldwide in order to generate additional evidence concerning CAP intake and its effects on health.

Several studies support the role of CAP in energy metabolism and fat oxidation, although the magnitude of these effects is modest [44]. In this research, a positive association was evidenced: the greater the consumption of CAP, the higher the level of adiposity. This is consistent with the results of previous epidemiological studies showing positive associations between chili intake and obesity measurements based on BMI, HC and BAI [19,20,21,22,23,24]. However, an important point of discussion is that the previous studies analyzing the associations between CAP intake and obesity status focused on the consumption levels of chili or spicy condiments, but CAP quantification was not necessarily estimated. Therefore, it is difficult to compare our results with those reported elsewhere, since the CAP content varies depending on the type of chili, the frequency of consumption and the form of culinary preparation [4,33]. Moreover, eating, cultural, genetic and social differences between regions should be considered [42]. Discerning the differences between the analyses of chili and CAP intake levels is important since chilies contain other nutritional components, such as fiber, vitamin C and beta-carotene, which can influence energy balance levels. Although the precise mechanisms have not been completely elucidated, plausible explanations for the positive relationship between chili consumption and obesity predisposition have been postulated, including cooking procedures (i.e., oil is added to make chili sauces, increasing energy input levels), and the increased intake of carbohydrate-rich foods (i.e., bread and sugary drinks) to relieve the burning sensation of pungent foods [21]. While statistical analyses were appropriately adjusted, our results show an increase in appetite as the CAP intake increases, which does not completely discard its influence on adiposity outcomes. Furthermore, the screening of other variables, such as hunger and cravings, should be addressed. Given the inconsistent evidence presented to date, additional investigations regarding the effects of CAP intake levels on adiposity are warranted, including longitudinal epidemiological studies and long-term randomized trials.

In animal models, dietary CAP triggered the browning of white adipose tissue in mice, counteracting obesity through the activation of TRPV1 channel-dependent mechanisms [45]. Of note, CAP intake exerted anti-obesity effects on high-fat diet (HFD) mice through alterations in gut microbiota populations (i.e., increases in the numbers of *Allobaculum*, *Coprococcus*, *Bacteroides*, *Prevotella*, *S24-7*, *Akkermansia* and *Odoribacter*, as well as reductions in *Sutterella*, *Helicobacter*, *Escherichia* and *Desulfovibrio*) and the production of intestinal short-chain fatty acid concentrations, including acetate and propionate [46]. Correspondingly, CAP administration in HFD-mice induced a significant reduction in weight gain levels and the improvement of glucose tolerance, which was related to a higher abundance of gut *Akkermansia muciniphila* [47]. In addition, the topical application of CAP to obese mice limited fat accumulation in adipose tissues by affecting adipokine levels involving adiponectin, PPAR-alpha and gamma, visfatin, adipsin, tumor necrosis factor-α and interleukin 6 [48].

In humans, a reduced TRPV1 expression in visceral adipose tissue from obese individuals was reported, which was accompanied by reduced CAP-induced calcium influx, highlighting the potential to activate TRPV1 channels to prevent obesity [49]. In addition, dietary CAP consumption showed beneficial effects for weight management in humans by reducing energy intake and triggering brown adipose tissue activity, leading to increased energy expenditure via non-shivering thermogenesis [50].

In addition to adiposity, the implication of CAP in other metabolic processes, such as liver function, has been reported, but the available research has mainly been conducted on animal models [51,52,53]. For example, chronic dietary capsaicin intake prevented non-alcoholic fatty liver disease (NAFLD) in mice through TRPV1-mediated peroxisome proliferator-activated receptor δ (PPARδ) activation [51]. Correspondingly, the topical application of capsaicin suppressed hepatic lipid accumulation through the upregulation of β-oxidation and de novo lipogenesis in diet-induced NAFLD mice [52]. Overall, it seems that the protective role of CAP against NAFLD is related to the amelioration of lipid alterations, oxidative stress, inflammation and fibrogenesis, as well as improvements in gut dysbiosis and increases in bile acid production [53]. Nevertheless, the studies on humans are scarce, to date. The findings of the present study show a positive association between CAP intake and markers of fatty liver, being apparently the first study on humans to report this association. This finding may be related to the increased adiposity observed in CAP consumers, which can favor abnormal fat deposition in the liver, but it seems not to be related to diet since no nutritional differences between CAP groups are observed. Although no differences between CAP groups were detected for the FLI, HIS and LDH variables in the ANOVA tests, these were positively correlated, which may have been due to the use of CAP consumption as a linear variable as well as the incorporation of covariates (age, sex, appetite, energy and macronutrient intakes) in the statistical settings. However, more studies on humans are required to confirm our results. Additionally, both HSI [54] and FLI [55] were valid, simple and efficient screening tools for the diagnosis of incipient liver steatosis, which supported their use in our investigation. Comparable observational studies have implemented these proxies to evaluate the associations between dietary factors (i.e., intake of specific nutrients and foods) and NAFLD [56], as well as to explore their roles as predictors of individual sensitivity to lifestyle changes comprising an adherence to an energy-restricted Mediterranean regime and physical activity [57].

The strengths of this study include the considerable sample of subjects analyzed, with a statistical power (80%) value similar to previous comparable studies on the Mexican population analyzing the associations between dietary intake and metabolic risk [31,32]. In order to reduce the bias in our results, statistical tests were adjusted for confounding factors, including the intake levels of total calories, fat, protein, carbohydrates and appetite, which could mask the associations between CAP consumption and adiposity/fatty liver status. The application of FFQ presented some limitations, such as systematic errors and biases in the estimates presenting inaccuracies due to the incomplete listing of all possible foods consumed, errors in frequency and usual serving sizes and dependence on the respondent’s ability to provide an answer question concerning eating according to social expectations, thus resulting in over- or under-estimations of certain food consumption levels [58]. To address these biases, we used a validated questionnaire to determine CAP consumption levels, which was based on Mexican chilies (fresh, dried or as condiments in prepared dishes) typically consumed throughout the Mexican territory. This instrument was supervised by trained nutritionists and included pre-determined portion sizes to improve the accuracy of the estimates. Moreover, this tool has been previously used to explore the associations between chili and CAP consumption levels with the risk of developing gastric cancer in Mexicans [4,33], reinforcing its application in our study to estimate CAP intake levels. On the other hand, some limitations comprise included the fact that this was a cross-sectional study, and therefore causal relationships could not be inferred. Moreover, type-I and -2 statistical errors could not be completely discarded. Thus, additional epidemiological studies in other regions of Mexico are needed, as well as experimental trials analyzing the mechanisms through which CAP affects adiposity and liver status. Furthermore, it is important to analyze the role of genetics associated with CAP tolerance and how this impacts health status. For instance, it has been reported that single nucleotide polymorphisms (SNPs) in the *TRPV1* gene influence some CAP-induced sensory changes in apparently healthy subjects, such as a burning sensation [59] and cough sensitivity [60]. Moreover, differential bitterness in CAP was associated with SNPs in the taste 2-receptor member 38 (*TAS2R38*) gene [61]. Additionally, variations in genes involved in the metabolism of CAP, such as cytochrome P450 family 2 subfamily E member 1 (*CYP2E1*), cytochrome P450 family 2 subfamily C member 9 (*CYP2C9*) and cytochrome P450 family 3 subfamily A member 4 (*CYP3A4*), may explain the heterogeneous responses to CAP intake [62].

## 5. Conclusions

Although caution should be exercised, the results of this study suggest modest but positive associations between dietary CAP consumption and markers of adiposity and fatty liver in a Mexican adult population. Further studies are needed to confirm our results as well as verify the status of other metabolic indicators based on CAP intake capacity.

## Figures and Tables

**Figure 1 healthcare-11-03001-f001:**
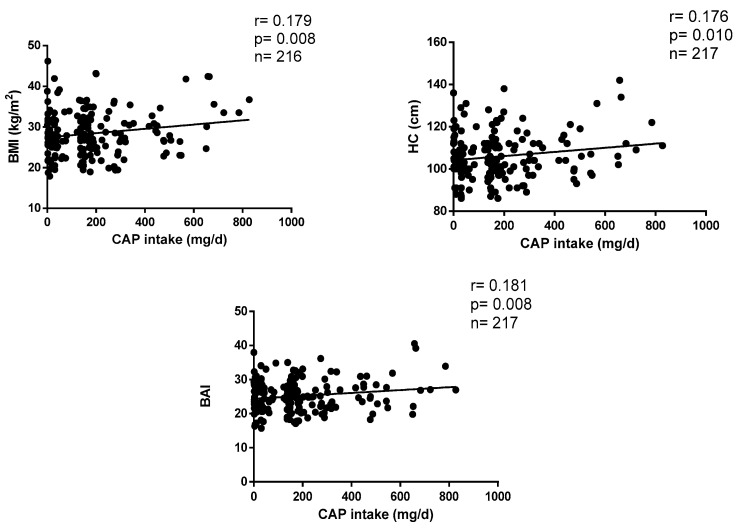
Correlations between CAP consumption and anthropometric parameters.

**Figure 2 healthcare-11-03001-f002:**
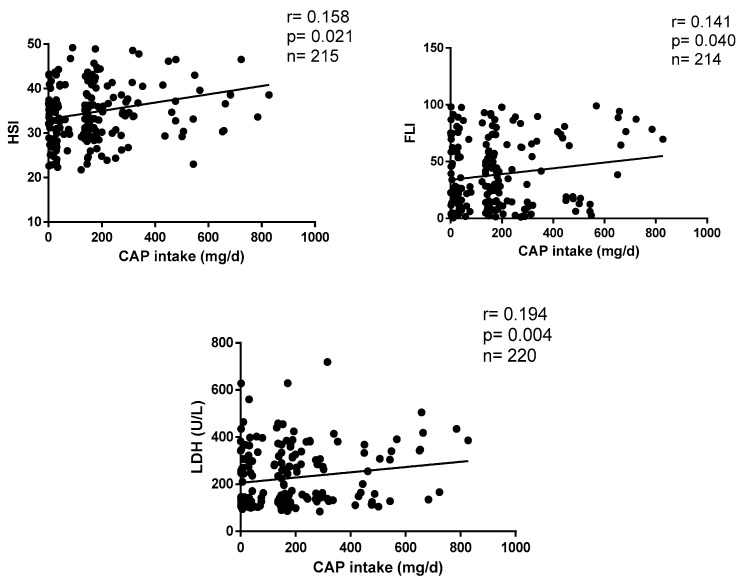
Correlations between CAP consumption and liver parameters.

**Table 1 healthcare-11-03001-t001:** Demographic, anthropometric and clinical characteristics of the total population stratified by CAP consumption.

Variable	Occasional CAP Consumers (*n* = 37)	Daily CAP Consumers, 0–100 mg (*n* = 66)	Daily CAP Consumers, 100–400 mg (*n* = 93)	Daily CAP Consumers, 400–900 mg (*n* = 25)	*p*-Value
Age (years)	39.0 ± 13.72	38.8 ± 12.18	36.8 ± 12.18	35.2 ± 12.26	0.499
Sex (F/M)	19/15	115/69	115/69	115/69	0.891
BMI (kg/m^2^)	27.4 ± 5.18	27.9 ± 5.70	28.1 ± 5.13	30.5 ± 5.92	0.147
Body fat (%)	33.8 ± 8.55	33.9 ± 10.3	34.5 ± 8.25	39.3 ± 8.74	0.260
WC (cm)	89.2 ± 16.2	89.5 ± 13.6	89.3 ± 14.1	95.3 ± 15.5	0.299
HC (cm)	103 ± 9.87	105 ± 10.6	104 ± 9.87	110 ± 12.3	0.074
WHR	0.85 ± 0.10	0.84 ± 0.08	0.85 ± 0.09	0.86 ± 0.10	0.875
SBP (mmHg)	121 ± 17.4	118 ± 21.5	121 ± 16.0	120 ± 15.5	0.817
DBP (mmHg)	80.1 ± 10.1	78.1 ± 10.3	78.6 ± 9.95	78.0 ± 9.93	0.779
BAI	22.9 ± 7.59	24.9 ± 4.62	24.8 ± 4.20	26.9 ± 5.51	**0.033 ***
VAI	3.99 ± 2.90	4.17 ± 2.87	4.73 ± 5.47	4.09 ± 2.77	0.741

Values are presented as means ± standard deviations. Sex is represented as number of cases. Bold numbers indicate *p* < 0.05. F: female; M: male; BMI: body mass index; WC: waist circumference; HC: hip circumference; WHR: waist to hip ratio; SBP: systolic blood pressure; DBP: diastolic blood pressure; BAI: body adiposity index; VAI: visceral adiposity index. * Post hoc tests: occasional CAP consumers vs. daily CAP consumers, 400–900 mg, *p* = 0.020.

**Table 2 healthcare-11-03001-t002:** Nutritional characteristics of the total population stratified by CAP consumption.

Variable	Occasional CAP Consumers (*n* = 37)	Daily CAP Consumers, 0–100 mg (*n* = 66)	Daily CAP Consumers, 100–400 mg (*n* = 93)	Daily CAP Consumers, 400–900 mg (*n* = 25)	*p*-Value
Appetite	10.1 ± 4.20	11.5 ± 2.22	11.5 ± 2.35	12.0 ± 2.29	**0.029 ***
Total calories	1898 ± 831	2205 ± 772	1975 ± 693	1995 ± 755	0.196
Proteins (%)	19.6 ± 9.89	18.3 ± 4.16	18.6 ± 4.06	20.8 ± 5.17	0.250
Fat (%)	38.4 ± 8.51	38.8 ± 8.36	37.0 ± 8.10	36.9 ± 6.9	0.533
Carbohydrates (%)	41.7 ± 12.4	41.8 ± 9.12	43.4 ± 9.30	41.8 ± 8.01	0.720

Values are presented as means ± standard deviations. Bold numbers indicate *p* < 0.05. * Post hoc tests: occasional CAP consumers vs. daily CAP consumers, 100–400 mg, *p* = 0.023; occasional CAP consumers vs. daily CAP consumers, 400–900 mg, *p* = 0.033.

**Table 3 healthcare-11-03001-t003:** Metabolic characteristics of the total population stratified by CAP consumption.

Variable	Occasional CAP Consumers (*n* = 37)	Daily CAP Consumers, 0–100 mg (*n* = 66)	Daily CAP Consumers, 100–400 mg (*n* = 93)	Daily CAP Consumers, 400–900 mg (*n* = 25)	*p*-Value
Blood glucose (mg/dL)	95.6 ± 21.4	94.9 ± 12.5	95.0 ± 11.6	94.4 ± 11.4	0.989
Total cholesterol (mg/dL)	191 ± 39	191 ± 36	190 ± 37	195 ± 46	0.947
HDL-c (mg/dL)	43.6 ± 11.1	47.2 ± 13.1	45.0 ± 14.0	46.6 ± 12.9	0.548
LDL-c (mg/dL)	128 ± 36	123 ± 29	123 ± 36	127 ± 38	0.832
Triglycerides (mg/dL)	102 ± 55	105 ± 60	108 ± 64	109 ± 61	0.954
Urea (mg/dL)	23.5 ± 14.4	24.7 ± 11.9	24.2 ± 13.1	24.7 ± 12.7	0.971
Total proteins (g/dL)	7.38 ± 0.49	7.37 ± 0.44	7.52 ± 0.46	7.49 ± 0.46	0.155
Albumin (g/dL)	4.04 ± 0.19	4.00 ± 0.17	4.03 ± 0.19	4.01 ± 0.20	0.725
Globulins	3.33 ± 0.048	3.37 ± 0.46	3.49 ± 0.47	3.45 ± 0.45	0.232
Creatinine (mg/dL)	0.64 ± 0.18	0.70 ± 0.18	0.68 ± 0.20	0.65 ± 0.20	0.188
LDH (U/L)	179 ± 91	217 ± 125	225 ± 125	256 ± 124	0.087
AST (U/L)	44.0 ± 41.0	37.3 ± 15.5	36.2 ± 8.83	37.3 ± 10.1	0.241
ALT (U/L)	36.7 ± 59.7	28.0 ± 21.1	27.3 ± 17.2	31.4 ± 22.3	0.416
GGT (U/L)	19.3 ± 15.5	19.0 ± 15.3	1817 ± 11.1	20.1 ± 18.3	0.921
TB (µmol/L)	1.01 ± 0.33	1.22 ± 0.56	1.15 ± 0.62	1.20 ± 0.44	0.320
DB (µmol/L)	0.14 ± 0.06	0.17 ± 0.09	0.16 ± 0.07	0.18 ± 0.05	0.289
ALP (U/L)	150 ± 67	154 ± 57	163 ± 63	154 ± 64	0.692
Uric acid (mg/dL)	5.41 ± 1.89	5.27 ± 1.85	5.61 ± 1.99	5.45 ± 1.74	0.745
Total calcium (mg/dL)	9.28 ± 1.42	9.51 ± 0.50	9.65 ± 0.40	9.58 ± 0.42	0.064
Total phosphorus (mg/dL)	2.88 ± 0.41	2.98 ± 0.53	3.02 ± 0.57	2.76 ± 0.38	0.117
Total iron (µmol/L)	90.6 ± 33.6	84.6 ± 31.3	84.8 ± 36.4	94.3 ± 32.7	0.518
Creatine kinase (U/L)	245 ± 240	377 ± 696	236 ± 263	202 ± 198	0.156
Sodium (mmol/L)	141 ± 1.28	141 ± 1.22	141 ± 1.17	141 ± 1.03	0.931
Potassium (mmol/L)	4.04 ± 0.30	4.04 ± 0.24	4.04 ± 0.28	3.99 ± 0.24	0.708
Chlorine (mmol/L)	103 ± 1.05	103 ± 1.17	103 ± 3.99	103 ± 1.11	0.725
FLI	35.4 ± 31.2	36.3 ± 30.3	37.5 ± 29.4	48.1 ± 33.7	0.344
HSI	34.2 ± 7.39	33.9 ± 8.26	34.7 ± 6.89	37.6 ± 7.55	0.214
TyG	4.52 ± 0.28	4.52 ± 0.29	4.54 ± 0.28	4.55 ± 0.26	0.941

Values are presented as means ± standard deviations. HDL-c: high-density lipoprotein cholesterol; LDL-c: low-density lipoprotein cholesterol; LDH: lactic dehydrogenase, AST: aspartate aminotransferase; ALT: alanine aminotransferase; GGT: gamma glutamyl transpeptidase; TB: total bilirubin; DB: direct bilirubin; ALP: alkaline phosphatase; FLI: fatty liver index; HSI: hepatic steatosis index; TyG: triglyceride and glucose index.

**Table 4 healthcare-11-03001-t004:** Inflammatory characteristics of the total population stratified by CAP consumption.

Variable	CAP Non-Consumers (*n* = 37)	CAP Consumption 0–100 (*n* = 66)	CAP Consumption 100–400 (*n* = 93)	CAP Consumption 400–900 (*n* = 25)	*p*-Value
NLR	1.96 ± 0.85	1.84 ± 0.95	1.97 ± 1.02	2.44 ± 1.72	0.140
PLR	135 ± 44.8	132 ± 63.8	147 ± 64.8	137 ± 42.4	0.459
EBR	3.29 ± 5.19	4.80 ± 8.79	4.75 ± 8.04	7.90 ± 17.1	0.305
ELR	0.07 ± 0.07	0.08 ± 0.05	0.10 ± 0.09	0.08 ± 0.06	0.335
LMR	5.86 ± 2.05	5.97 ± 2.24	6.04 ± 2.29	6.32 ± 3.45	0.898

Values are presented as means ± standard deviations. NLR: neutrophil to lymphocyte ratio; PLR: platelet to lymphocyte ratio; EBR: eosinophil to basophil ratio; ELR: eosinophil to lymphocyte ratio; LMR: lymphocyte to monocyte ratio.

## Data Availability

The data presented in this study are available on request from the corresponding authors.

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
