# Peer review of "Dietary Intake of Capsaicin and Its Association with Markers of Body Adiposity and Fatty Liver in a Mexican Adult Population of Tijuana"

_healthcare, 2023, doi:10.3390/healthcare11223001_

Round 1

Reviewer 1 Report

Comments and Suggestions for Authors

The relationship between dietary intake of capsaicin and body adiposity and fatty liver in a Mexican adult population was investigated in this study. Although there were controversial results reported especially in Chinese adult population, the authors demonstrated capsaicin consumption positively correlated with body adiposity and fatty liver in a northwest Mexican adult population. This study is interesting and convincing. The references are appropriate.

The most of first paragraph of the Discussion should be moved to the Introduction.

In Tables 1, 2, 3 and 4 the authors simply divided subjects into two groups. According to the data in Figures 1 and 2, four groups (Non-consumers, consumption 0 – 100 mg/d, 100 – 400 mg/d and 400 -900 mg/d) were suggested to analysis.

Author Response

Please, see attached file.

Reviewer 2 Report

Comments and Suggestions for Authors

I have thoroughly reviewed this manuscript with keen interest in the research topic it covers, and the results are as follows.

1.      In the introduction section, the authors need to address the following points:

l  Provide an overview of prior epidemiological studies on the effects of chili and capsaicin, particularly those involving Mexican populations, to establish the context.

l  Present the usage rates of Mexican chili or capsaicin to give a clear understanding of their prevalence in the Mexican diet.

l  Highlight the prevalence of body adiposity and fatty livers in Mexicans to emphasize the health issues of concern.

l  Discuss health problems related to body adiposity and fatty liver in Mexicans, and describe the healthcare measures that have been implemented or are currently in progress, either separately or at a national level, to tackle these issues effectively.

2.      In the materials and methods section, it is imperative to provide the following:

l  Recruitment Process and Participant Selection Method: A thorough explanation of the recruitment process, participant selection methodology, the basis for choosing the age range (18 to 65 years old), and information regarding dropout rates.

l  Criteria for Selection and Exclusion of Research Participants: A clear and theoretically sound description of the criteria used to select and exclude research participants, including the reasoning behind these criteria.

l  Personnel and Location for Anatomical Measurements: Details about who conducted the anatomical measurements and where these measurements were performed.

3.      In the results section, it is essential to provide clear criteria for the CAP (capsaicin) consumers group, including details about the CAP intake period and intake levels.

l  In Table 1, the authors state that "an individual who consumes capsaicin (CAP) every day exhibits higher levels of HC (p=0.032) and a greater BAI (p=0.022) compared to those who do not consume CAP." However, the rationale for attributing these differences to CAP consumption is not clearly explained.

l  Furthermore, in "Figure 2," the authors report that daily capsaicin (CAP) intake shows a positive correlation with HSI (r=0.158, p=0.004), FLI (r=0.141, p=0.003), and LDH (r=0.194, p=0.016) after statistical adjustments. However, Table 3 notes that there were no significant differences between the two groups except for total bilirubin (TB) and creatinine levels among the metabolic indicators. This inconsistency necessitates further explanation and discussion to clarify the results.

4.      In the discussion section, it is essential to carefully consider the following points and content being presented:

l  In the discussion, the authors claim, "The results of this study demonstrate a positive association between capsaicin (CAP) intake and fatty liver markers, marking the first study to report such an association in humans." However, Table 3 reveals that no significant differences in metabolic indicators were observed between the two groups (those with or without CAP intake) except for the creatinine index and total bilirubin (TB). Additionally, Figure 2 indicates a positive relationship between CAP intake and fatty liver markers. To provide a comprehensive analysis, it is crucial to conduct statistical analysis for the association between the group that did not consume CAP and the fatty liver markers as well.

l  In the discussion, the authors have predominantly relied on animal research results as evidence for the outcomes of this study, with limited reference to prior research on human subjects. However, several previous studies have provided theoretical evidence that white fat is more prevalent than brown fat in obese individuals. Therefore, it is essential to discuss these findings within the context of prior human studies.

5.      In the conclusion, the authors have concluded that this study suggests a positive association between dietary capsaicin (CAP) intake and fat and fatty liver indicators in the Mexican adult population. However, to bolster this conclusion, further verification of the association with metabolic indicators based on CAP intake capacity is required.

Author Response

Please, see attached file.

Reviewer 3 Report

Comments and Suggestions for Authors

The manuscript follows on two studies in China that evaluated relationship between CAP intake and body adiposity measures in Chinese population and gave conflicting results. Here, the authors evaluate these associations among individuals living in Tijuana in Mexico. They found positive association between CAP consumption BMI, HC, BAI, etc. This manuscript is of interest, but some improvements need to be done to strengthen its message.

Title: add “Mexican population of Tijuana”; the current title is misleading, as only population of one Mexican city was evaluated.

Previously only two studies in China evaluated the association between capsaicin consumption and markers of adiposity and inulin resistance and the y reach conflicting conclusions. No discussion why that could have happened. Were there any methodological issues in both of them that this study attempted to correct? Furthermore, a simple query of Pubmed shows that there were many studies evaluating the effects of CAP on energy expenditure and satiation. Please add a short review of these studies to the introduction. Similarly, the description of other studies relating CAP intake with adiposity should be moved from the discussion to the introduction. Describe also in more details previous “comparable” studies in Mexican population. Were the conclusions consistent with yours?

No information on how the participants were recruited? Were they randomly included? Was there any method of stratification used?

Are any participants taking medications for diabetes or decreasing blood pressure, cholesterol, etc.? Are the healthy by self-report or medical evaluation?

Describe shortly BAI and VAI, as some of the readers may be not acquainted with these measures. Why were they calculated in this study in the first place?

24 h recall and a validated in Mexican population FFQ to estimate nutritional intake of each participant. They should allow to calculate each participant’s caloric intake.  

Table 1: change naming: CAP non-consumers à“daily CAP consumers” and e.g., “occasional CAP consumers”. Feel free to come up with a more relevant name of the latter variable.

Huge variability in WHR among the CAP-non-consumers. Please provide a histogram of WHR distribution. Is there a group of morbidly obese individuals among CAP-non-consumers? If not, what is driving the variability so high? Please note that HWR variability is relatively tight among CAP consumers.

Discussion:

Summary of findings should be in the first paragraph of the discussion.

Author Response

Please, see attached file.

Round 2

Reviewer 2 Report

Comments and Suggestions for Authors

This manuscript has been appropriately modified in accordance with the reviewer's requests. However, the following issues have not been resolved:

1.       According to the study results, a significant difference exists between CAP intake and variables such as BAI and appetite based on CAP dose. In the discussion, the authors must provide a more detailed description of the influencing factors for these findings.

2.       The research results presented in Figures 1 and 2 indicate that the correlation coefficient (r) values are both less than 0.2. Generally, a correlation coefficient of approximately 0.2, in absolute value, does not guarantee a correlation and is interpreted as a weak (ambiguous) correlation, requiring further research. However, in the discussion and conclusion sections, the authors state, "The findings of this study suggest positive associations between CAP intake and markers of adiposity (BMI, BAI, and HC) and fatty liver (HSI, FLI, and LDH) in the adult Mexican population of Tijuana." The authors must delve deeper into this discussion.

Author Response

Please, see attached file. 
